# Acute Effect of a Simultaneous Exercise and Cognitive Task on Executive Functions and Prefrontal Cortex Oxygenation in Healthy Older Adults

**DOI:** 10.3390/brainsci12040455

**Published:** 2022-03-28

**Authors:** Manon Pellegrini-Laplagne, Olivier Dupuy, Philippe Sosner, Laurent Bosquet

**Affiliations:** 1Laboratoire MOVE (UR 20296), Faculté des Sciences du Sport, Université de Poitiers, TSA 31113, CEDEX 9, 96073 Poitiers, France; manon.pellegrini.laplagne@gmail.com (M.P.-L.); psosner@monstade.fr (P.S.); laurent.bosquet@univ-poitiers.fr (L.B.); 2Société Rev’Lim, 87000 Limoges, France; 3Faculty of Medicine, School of Kinesiology and Physical Activity Sciences (EKSAP), University of Montreal, Montreal, QC HC3 3J7, Canada; 4Centre Médico-Sportif MON STADE, 75013 Paris, France; 5AP-HP Hôtel-Dieu, Centre de Diagnostic et de Thérapeutique, 75004 Paris, France

**Keywords:** physical exercise, cognitive exercise, simultaneous training, healthy aging, executive function, cerebral oxygenation

## Abstract

The rapid increase in population aging and associated age-related cognitive decline requires identifying innovative and effective methods to prevent it. To manage this socio-economic challenge, physical, cognitive, and combined stimulations are proposed. The superiority of simultaneous training compared to passive control and physical training alone seems to be an efficient method, but very few studies assess the acute effect on executive function. This study aimed to investigate the acute effect of simultaneous physical and cognitive exercise on executive functions in healthy older adults, in comparison with either training alone. Seventeen healthy older adults performed three experimental conditions in randomized order: physical exercise, cognitive exercise, and simultaneous physical and cognitive exercise. The protocol involved a 30 min exercise duration at 60% of theoretical maximal heart rate or 30 min of cognitive exercise or both. Executive functions measured by the Stroop task and pre-frontal cortex oxygenation were assessed before and after the intervention. We found a main effect of time on executive function and all experimental condition seems to improve inhibition and flexibility scores (<0.05). We also found a decrease in cerebral oxygenation (Δ[HbO_2_]) in both hemispheres after each intervention in all cognitive performance assessed (*p* < 0.05). Simultaneous physical and cognitive exercise is as effective a method as either physical or cognitive exercise alone for improving executive function. The results of this study may have important clinical repercussions by allowing to optimize the interventions designed to maintain the cognitive health of older adults since simultaneous provide a time-efficient strategy to improve cognitive performance in older adults.

## 1. Introduction

Age-related decline in cognitive performance, such as a decrease in processing speed, attention, and executive functions can be associated with cognitive disorders such as dementia [1,2,3,4]. Considering the demographic evolution in many countries, the increase in the incidence of cognitive impairment in the near future is expected to double within the next 20 years [5]. Consequently, there is a need to identify innovative and effective treatments to prevent, reduce, and delay functional impacts of normal age-related cognitive decline on the daily life of healthy older adults. In this context, one of the main challenges consists of developing efficient strategies to strengthen the “cognitive reserve” of the elderly [6]. Fortunately, drug-free interventions aimed at mitigating cognitive decline are attracting great research and societal interest. To date, three strategies have been used to prevent cognitive decline and dementia: (i) cognitive training, (ii) exercise training, and (iii) simultaneous cognitive and exercise training [7]. Due to the positive impact of separate cognitive and physical training, Kraft put forward the hypothesis that simultaneous cognitive and physical training could induce a better cognitive improvement than physical training or cognitive training alone [8]. This hypothesis has been supported by a recent meta-analysis by Gavelin et al., 2021 who reported that combined cognitive and exercise training was more efficient to improve cognition performances than each stimulation alone [9].

For several years, a growing body of literature has investigated the acute effect of a single bout of aerobic exercise on cognitive functions. Although this effect has been mostly described in healthy young adults [10,11,12,13,14], the interest in the specific response of older adults is more recent [15,16,17]. The main observation is a positive impact of acute exercise on executive functions [16], the magnitude of the effect being larger in older vs. younger adults [17]. Surprisingly, the acute effect of a task combining simultaneously a physical and a cognitive exercise has received very little attention. To the best of our knowledge, the work by Ji et al., 2019 [18] is the single one that experimental session involving a cognitive exercise, a physical exercise, a combined cognitive and physical exercise, and a reading exercise that served as a control condition. Pre-frontal cortex oxygenation was measured during a Stroop task test that was performed before and after each exercise. The authors failed to report greater effect for combined modality and showed an improvement in executive performance after the physical exercise and the combined cognitive and physical exercise, but not after the cognitive exercise and the control condition. Interestingly, they also reported greater oxygenation of the pre-frontal cortex after the combined physical and cognitive exercise in comparison with the other conditions in post-exercise condition. These results provide interesting insights into the adaptation process of the pre-frontal cortex to an acute bout of combined cognitive and physical exercise, but they still need to be confirmed [18]. The lack of superiority of simultaneous exercise on cognitive functions in the elderly reported by Ji et al., 2019 [18] can potentially be explained by the cognitive exercise used. Indeed, these authors used verbal fluency as a cognitive exercise and tested its effect on Stroop performance, particularly on flexibility. It is known to date that cognitive training has a positive effect on the stimulated cognitive function only and the transfer to another cognitive process is difficult [19]. It is now important to test the effect of multiple cognitive exercises by exercising several cognitive and executive functions.

The aim of this study was therefore to investigate the acute effect of simultaneous physical and cognitive exercise using multiple stimulated cognitive functions on executive functions in healthy older adults in comparison with either exercise alone. Based on Kraft theory [8], we hypothesized that simultaneously combined cognitive and physical exercise would be effective in improving executive performance, that the magnitude of improvement would be greater than either exercise mode alone, and that this effect would be mediated by a greater increase in pre-frontal cortex oxygenation.

## 2. Methods

### 2.1. Population

In total, 17 men (*n* = 5) and women (*n* = 12) aged between 57 and 69 years old were recruited to participate in this study. Their characteristics are presented in Table 1. None of them suffered from cognitive impairment (as defined as a MOCA score inferior or equal to 24) or from cardiovascular, metabolic, neurological, or psychiatric diseases. Participants who have been prescribed with pharmacological treatment that could modify cardiovascular or neuromuscular functions were excluded. The protocol was reviewed and approved by a national ethics committee for non-interventional research (CERSTAPS # 17 November 2017) and was conducted by recognized ethical standards and national/international laws. All participants signed a written statement of informed consent.

### 2.2. Study Design

Participants completed four different experimental sessions. The first session consisted of familiarization with the computerized modified Stroop task test and a submaximal intensity exercise test to determine the power necessary to achieve 60% of theoretical maximal heart rate (*t*HRmax). In the following sessions, participants performed the computerized modified Stroop task before and after a 30 min bout of either cognitive (CE), physical (PE), or combined exercise (SE), in random order. The period separating each of these three experimental visits was one week. Each participant was tested each time at the same time of day. Pre-frontal cortex oxygenation was measured continuously during the cognitive test. Considering the observation by Chang et al., 2012 that greater benefits of physical exercise on cognitive performance were observed 15 min after exercise cessation, the Stroop task was performed 10 min after each condition cessation [20]. The study design is presented in Figure 1.

#### 2.2.1. Submaximal Intensity Exercise Test

This test was performed on a stationary bicycle (Monark LC6, Monark Exercise AB, Vansbro, Sweden). Theoretical maximal heart rate was determined by the formula proposed by Gellish and colleagues [21]:*t*HR_max_ = 207 − 0.7 ∗ age(1)

The test began at 30 W and increased by 20 W every two minutes until the participant reached 60% of *t*HR_max_. Participants had to pedal at 60 rotations per minute (rpm). Oxygen uptake (VO_2_ mL·kg^−1^·min^−1^) was determined continuously on a 30-s basis using a portable cardiopulmonary exercise testing system (Metalyser Cortex 3B, CORTEX Biophysik GmbH, Germany). Gas analyzers were calibrated before each test using ambient air and a gas mixture of known concentrations (15% O_2_ and 5% CO_2_). The turbine was calibrated before each test using a 3-l syringe at several flow rates. The highest VO_2_ over a 30-s period during the last stage was considered as the oxygen uptake at 60% of *t*HR_max_ (VO_2 peak(60%),_ mL·kg^−1^·min^−1^). Heart rate was measured continuously using a heart rate monitor (Polar RS800 cx, Polar Electro, Kempele, Finland).

#### 2.2.2. Physical Exercise

Participants completed a 30 min session of stationary bicycle at 60% of their *t*HR_max_, using a specific device (Velo-Cognitif, REV’LIM, Limoges, France, Figure 2). The corresponding power was estimated during the submaximal intensity exercise test. Heart rate was recorded continuously to adjust power output during the session. The choice of exercise intensity of 60% of *t*HR_max_ is based on some observations suggesting that cognitive performance was improved at this intensity [22] and fulfils the recommendations by Lauenroth et al., 2016 [23].

#### 2.2.3. Cognitive Exercise

Participants completed a 30 min session of cognitive games in PRESCO (HappyNeuron, Grenade Sur Garonne, France). This software was already used by [24]. The cognitive exercise consisted of 4 different games (among 32 possible games), which were the same for all participants but with a level of difficulty adapted to their cognitive capacities. These 4 games were used to train mainly executive functions. The testing order, the objective, and the description of each cognitive game are presented in Table 2. The procedure of the cognitive exercise session is illustrated in Figure 3. Each cognitive game had an approximate duration of 7 min, and 30 s of rest was used between each game. Cognitive games were presented on a touch screen, and subjects were asked to perform these games by responding on the touch screen (Figure 2).

#### 2.2.4. Simultaneously Combined Physical and Cognitive Exercise

Participants performed a 30 min session of simultaneously combined physical and cognitive exercise, using a specific device (Velo-Cognitif, REV’LIM, Limoges, France, Figure 2) and the same modalities as previously described for the cognitive exercise and physical exercise. Generally, the cognitive exercise started after 30 s of the beginning of pedaling.

### 2.3. Neuropsychological Assessment

#### 2.3.1. Global Cognitive Test

The Montreal Cognitive Assessment test (MOCA) was used to evaluate global intellectual efficiency. Briefly, this test consists of a 30-point test divided into 8 parts and 14 subtests. It is a very complete test because it targets most cognitive functions: visuospatial skills, executive functions, attention, working memory, short-term memory, delayed recall, and language. A score higher than 26 is a normal score, and below this, people have a mild cognitive impairment [25]. The dependent variable was the total score.

#### 2.3.2. Executive Functions

The Computerized Modified Stroop task was used to assess executive functions. The test used in this study is based on the Modified Stroop Color Test and included three experimental conditions: naming, inhibiting, and switching. This task was already used in several articles [26,27,28,29] with young and older people. Each block lasted between 2–4 min and was interspersed with 60-s resting blocks. Overall, there were 3 experimental task blocks (1 naming, 1 inhibition, and 1 switching) and 2 resting blocks, for a total length between 8 and 14 min. In total, there were 60 Naming trials (Block 1), 60 Inhibition trials (Block 2), and 60 Switching trials (Block 3). All trials began with a fixation cross (or square for switching condition) for 1.5 s, and all visual stimuli appeared in the center of the computer screen for 2.5s. Participants responded with two fingers (index and major finger) from each of their hands on an AZERTY keyboard. In the Naming block, participants were presented with a visual stimulus of the name of colors (RED/BLUE/GREEN/YELLOW) in French presented in the color that is congruent with the word (i.e., RED presented in red ink). Participants were asked to identify the color of the ink with a button press. In the Inhibition block, each stimulus consisted of a color word (RED/BLUE/GREEN/YELLOW) printed in the incongruent ink color (e.g., the word RED was presented in blue ink). Participants were asked to identify the color of the ink (e.g., blue). In the Switching block, in 25% of the trials, a square replaced the fixation cross. When this occurred, participants were instructed to read the word instead of identifying the color of the ink (e.g., RED). As such, within the Switching block, there were both inhibition trials in which the participant had to inhibit their reading of the word and correctly identify the color of the ink, and there were switch trials in which the participant had to switch their response mode to read the word instead of identifying the color of the ink when a square appeared before the word presented. Visual feedback on performance was presented after each trial. A practice session was completed before the acquisition run to ensure the participants understood the task. The practice consisted of a shorter version of the task. Dependent variables were reaction times (ms) and the number of errors committed (%). We also calculated two scores, the first being the inhibition score, which is the result of the inhibition block’s score minus naming blocks’ performances on number or error, and corresponds to pure inhibition capacities. The second, flexibility cost, is equal to the results of flexibility block’s score minus inhibition blocks’ performances. The order of handover of executive blocks was counterbalanced between subjects. The Stroop task is presented in Figure 1.

### 2.4. Prefrontal Cortex Oxygenation Measurement

Cerebral oxygenation and, more precisely, the concentration changes in [HbO_2_] (Δ[HbO_2_]), HHb (Δ[HHb] and Δ[THb]) were recorded during the Stroop test with the OxyMon fNIRS system (Artinis Medical Systems, Elst, Netherlands). This system utilizes near-infrared light, which penetrates the skull and brain but is absorbed by hemoglobin (Hb) chromophores in capillary, arteriolar, and venular beds (Ferrari & Quaresima, 2012) [30]. The light was transmitted with two wavelengths, 764 and 857 nm, and data were sampled with a frequency of 10 Hz. This procedure measures relative changes in [HbO_2_] and [HHb] using the modified Beer–Lambert law. This law takes into account the differential pathlength factor (DPF), which is determined using the following formula: DPF (λ = 807 nm, A) = 4.99 + 0.067 × (age 0.814). In our study, the DPF ranged from 5.69 to 6.60. The region of interest is the Fp1 and Fp2 of the prefrontal cortex (PFC). The patch used in this study is presented in Figure 1 and used eight optical channels, comprising four emitters and four receptors, covering the right and left DLPFC and ventrolateral PFC (VLPFC) (Brodmann areas, BAs 9/46 and 47/45/44), which were located using the 10/20 international system. The distance between each emitter and receptor was 3.5 cm. The sensors were shielded from ambient light with a black cloth. Oxysoft version 3.0 (Artinis Medical Systems, Elst, Netherlands) was used for data collection. Initially resting PFC oxygenation was acquired in a seated position for 2 min before each Stroop task (before and after experimental conditions). Because continuous-wave technology does not allow quantifying absolute concentration due to the incapacity of measuring optical path lengths, the mean of [HbO_2_], [HHb], and [THb] during the total duration of each block of the Stroop were compared to the minute preceding each block. Our participants were asked to always face forward during the test, avoid making a sudden head movement, clenching their jaw, frowning, and other facial expressions. This procedure was already used by two different teams [26,27,28,31].

### 2.5. Statistical Analysis

Standard statistical methods were used for the calculation of means and standard deviations. Normal Gaussian distribution of the data was verified by the Shapiro–Wilks’s test and homoscedasticity by a modified Levene Test. Repeated measures ANOVA was used to test the interaction between time and exercise condition on cognitive performance. When the main effect was found, a Bonferroni post-hoc test was performed. All statistical analyses were made with SPSS 17.0, and all statistical analyses with a *p*-value < 0.05 were considered significant. Effects sizes (ES) were also calculated with Hedges’ g formula, previously described by Dupuy et al., 2015, and interpreted with Cohen’s scale, where EF ≤ 0.2 (trivial), >0.2 (small), >0.5 (moderate), and >0.8 (large) [26].

## 3. Results

### 3.1. Participants

The study included nineteen healthy adults aged between 57 and 69 years old. We removed two of them because of health considerations during the study. The final sample included 17 participants (12 women and 5 men). Their characteristics are presented in Table 1.

### 3.2. Cognitive Assessment

All cognitive results are presented in Table 3. We found no difference between condition and no interaction between time and condition in the congruent condition of the Stroop test (i.e., naming). In contrast, we found a main effect of time on the executive performance, including the inhibition reaction time (RT) (F_(1.45)_ = 7.4, *p* < 0.01, ES = −0.18) and the flexibility errors number (F_(1.44)_ = 6.65, *p* = 0.01, ES = −0.31) and RT (F_(1.45)_ = 16.5, *p* < 0.01, ES = −0.24)]. Regarding inhibition cost, we found no main effect of time [errors number (F_(1.44)_ = 1.37, *p* = 0.24); RT (F_(1.44)_ = 0.1, *p* = 0.80)]. In contrast, we found a main effect of time on the RT flexibility cost score (F_(1.44)_ = 4.17 *p* = 0.04, ES = −0.15) and no interaction between time and condition (*p* = 0.12). For all significant results, the effects sizes are calculated and presented in Figure 4. Based on the ES analysis (Figure 4), we found no effect of physical exercise (ES = −0.01) and cognitive exercise (ES = 0.11) on flexibility cost, whereas we found a moderate effect of combined cognitive and physical exercise (ES = −0.67).

### 3.3. PFC Oxygenation

All results for right and left hemispheres are presented in the Table 4. The results for the whole PFC are presented in the Figure 5. Repeated measured ANOVA revealed a main effect of time on Δ[HbO_2_] and on Δ[THB] during naming, inhibition and flexibility conditions of the Stroop test. More precisely, we found a decrease in [ΔHbO_2_] during naming (F_(1.44)_ = 10.6, *p* < 0.01, ES = −0.51), inhibition (F_(1.44)_ = 7.22, *p* = 0.01, ES = −0.44) and flexibility (F_(1.44)_ = 8.5, *p* < 0.01, ES = −0.45). Similarly, total Δ[THB] decreased during naming (F_(1.44)_ = 9.7, *p* < 0.01, ES = −0.58), inhibition (F_(1.44)_ = 7.2, *p* < 0.01, ES = −0.49) and flexibility (F_(1.44)_ = 4.28, *p* = 0.04, ES = −0.30). In contrast, total Δ[HHB] remained stable during each condition of the Stroop test [naming (F_(1.44)_ = 0.05, *p* = 0.81), inhibition (F_(1.44)_ = 0.19, *p* = 0.65) and flexibility (F_(1.44)_ = 0.31, *p* = 0.57)]. We found no differences between the left and right hemispheres.

## 4. Discussion

This study aimed to investigate the acute effect of simultaneous physical and cognitive exercise on executive functions in healthy older adults, in comparison with either exercise alone. Based on Kraft theory [8], we hypothesized that simultaneously combined cognitive and physical exercise would be effective in improving executive performance and that the magnitude of improvement would be greater than either exercise mode alone. We also hypothesized that this effect would be mediated by a greater increase in pre-frontal cortex oxygenation. Contrary to our hypotheses, we do not find either in our cognitive results, or in our cerebral oxygenation results, a superior effect of simultaneous exercise compared to physical or cognitive exercises alone. Our main findings were (1) an improvement in executive performance (i.e., inhibition and flexibility conditions of the Stroop task) after each exercise condition; (2), a larger decrease in flexibility cost (based on the ES) after simultaneous cognitive and physical exercise; and (3) a decrease in Δ[HbO_2_] and Δ[THb] after each exercise condition, both in the left and right PFC.

The effect of acute physical exercise on the cognitive performance of older adults has been summarized in systematic reviews and meta-analyses [15,16,17]. The Stroop task is probably the test that is the most widely used to assess the acute effect of physical exercise on cognitive performance. The results of the literature seem unclear with (i) specific positive effects on executive functions such as inhibition, (ii) effects only on processing speed, or (iii) even a general effect. It would seem that these contradictory effects are due to the multitude of exercise modalities used. Of the six studies identified by McSween’s systematic review (2019), exercise duration ranged from 10 min to 30 min using either cycling, walking, g, or stepping exercise modalities [15]. The Stroop task that we use contained three blocks that allow us to evaluate naming, inhibition, and flexibility functions. Our results showed that the exercise condition alone induces a beneficial effect on the reaction time of the inhibition and flexibility block, as well as the flexibility cost. However, we do not observe any effect on the processing speed on the inhibition cost. Our results validated the hypothesis that executive functions are more sensitive to acute physical exercise as shown by several authors [32,33,34]. Indeed, results from Abe et al., 2018, Hyodo et al., 2012, and Johnson et al., 2016 suggested an enhancement of the higher levels of executive functions assessed by the Stroop interference in the post-exercise condition, which is consistent with previous findings in young adults [35,36]. In addition, these authors reported no enhancement of information processing speed. Furthermore, our results validate the hypothesis that the most ‘demanding’ functions are more sensitive to the effect of acute exercise. Indeed, several researchers [37,38,39] suggest that more demanding tasks are likely to be more sensitive to the effects of physical exercise in comparison with automatic effortless tasks, which supports the results of Abe et al., 2018, Hyodo et al., 2012 and Johnson et al., 2016 [32,33,34].

The acute effects of cognitive exercise or simultaneous cognitive and physical exercise on cognitive function in the elderly have been little studied. To our knowledge, only Ji et al., 2019 [18] have studied this effect and reported a positive effect on inhibition and flexibility processes. Our results also confirm this effect. However, like Ji et al., 2019 [18] our results do not corroborate Kraft’s hypothesis [8] that double stimulation could have a greater effect than other conditions alone. It should be remembered that this hypothesis was formulated on long-term chronic effects and that we tested this hypothesis on acute effects after a single session. However, although our statistical approach did not report interaction x time on executive function, we can observe the largest effect (based on the ES) of the simultaneous stimulation on the flexibility cost compared to other stimulations (cognitive and physical alone). This result seems encouraging and confirms that this stimulation induces a beneficial effect and can be considered as a cognitive enhancement time efficiency strategy for elderly people.

The effects observed in this study on cerebral oxygenation showed a decrease in cerebral oxygenation during the three blocks of the Stroop task after three types of exercise. Indeed, compared to baseline, we observe a smaller increase in Δ[HbO_2_] and Δ[THB] during the naming, inhibition, and flexibility tasks. The effect of acute exercise on brain oxygenation during a cognitive task has been reviewed recently by Herold et al., 2018 [40]. Usually, in the vast majority of studies, greater brain oxygenation is observed during a cognitive task after exercise. However, this finding was observed in young subjects in most studies, which potentially explains the difference with the results obtained in our study. In addition, several studies that report a greater cerebral oxygenation during a cognitive task after a physical exercise uses a baseline before exercise and not just before each cognitive task. This procedure unfortunately does not allow us to assess only the cortical activity. However, Murata et al., 2015 [41] observed the same result after exercise of comparable intensity (i.e., 50% of VO_2max_). The oxygenated hemoglobin concentration quantified across the whole brain was lower after exercise, and this was the case for go trials and no-go trials. More precisely, the oxygenated hemoglobin concentration in the dorsolateral prefrontal cortex and the supplementary motor area was significantly lower after exercise. These authors hypothesized that brain activity is less important than before exercise for the go and the no-go tasks [41]. This hypothesis can be explained by the hypothesis provided by Audiffren et al., 2008 [42] who explains that acute exercise improves ‘arousal’ explaining the enhancement of cognitive performance in post-exercise conditions. The facilitating effect of exercise on cognitive performance may explain the lower cortical activity observed in our study [42]. This mechanism was observed after a period of chronic physical training where less brain activity was observed during a cognitive task. Coetsee et al., 2017 observed the same results with fNIRS techniques, reporting less HBO2 after a physical training program [43]. These results are in accordance with fMRI results from Volcker-Rehage et al., 2011 [44], who found less cortical activity after physical training. We could hypothesis that the brain is more efficient after acute exercise, and therefore requires less cortical and therefore metabolic and vascular activity. The reduced cerebral oxygenation after cognitive exercise may also reflect reduced cortical activity. This phenomenon could be explained by the fact that during cognitive exercise, neural circuits are stimulated and may benefit from a facilitating effect when stimulated later. This theory is only speculative and requires further work to validate.

Though this study had several strengths, it was not without limitations, and the interpretation of our results requires some caution. One limit of this article is the small sample of this study, which does not allow us to generalize our results on a large scale. Nevertheless, the calculation of the size of the effect using the Hedge formula allows us to appreciate our results from a clinical point of view. The second limitation is the lack of a control group. The presence of a control group would allow us to fully ensure that our cognitive results and cerebral oxygenation are not due to a test–retest effect. Nevertheless, the familiarization at the first visit and the practice trials before each block of the Stroop task enables us to control this effect. In addition, the study by Ji et al., using a Stroop task and a similar design to ours, did not report any change in the reaction time or error in the older control group.

## 5. Conclusions

Based on Kraft theory [8], we hypothesized that simultaneously combined cognitive and physical exercise would be more effective in improving executive performance than either exercise mode alone and that this effect would be mediated by a greater increase in pre-frontal cortex oxygenation. Our main findings were (1) an improvement in executive performance (i.e., inhibition and flexibility conditions of the Stroop task) after each exercise condition; (2) a larger decrease in flexibility cost (based on the ES) after combined cognitive and physical exercise; (3) a decrease in Δ[HbO_2_] and Δ[THB] after each exercise condition, both in the left and right PFC. Simultaneous physical and cognitive exercise is as effective a method as either physical or cognitive exercise alone for improving executive function and present no superiority in acute setting. The results of this study may have important clinical repercussions by allowing us to optimize the interventions designed to maintain the cognitive health of older adults since simultaneous provide a time-efficient strategy to improve cognitive performance in older adults.

## Figures and Tables

**Figure 1 brainsci-12-00455-f001:**
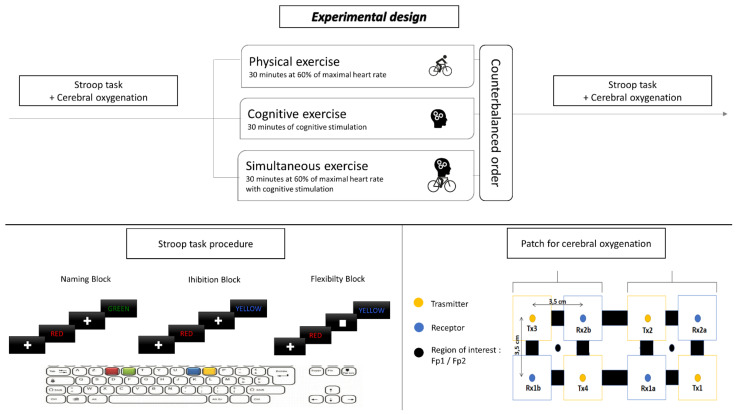
Illustration of experimental design, Stroop task and patch used for cerebral oxygenation.

**Figure 2 brainsci-12-00455-f002:**
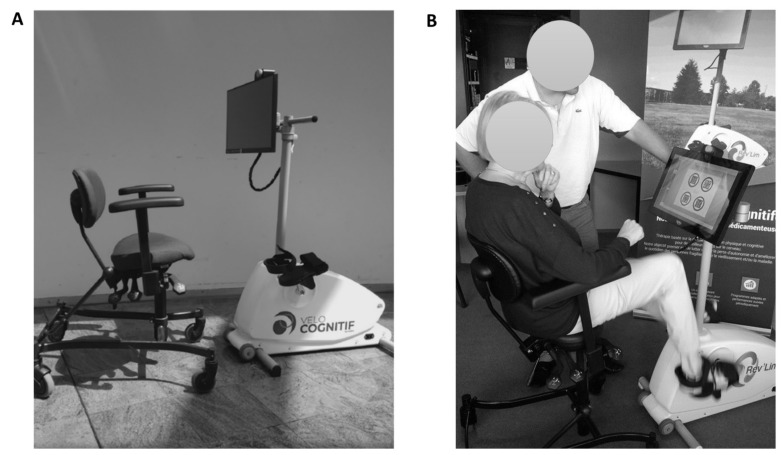
(**A**) Illustration of ‘Velo-Cognitif’, (**B**) example of participants during the simultaneous physical and cognitive exercises on ‘Velo-Cognitif’.

**Figure 3 brainsci-12-00455-f003:**
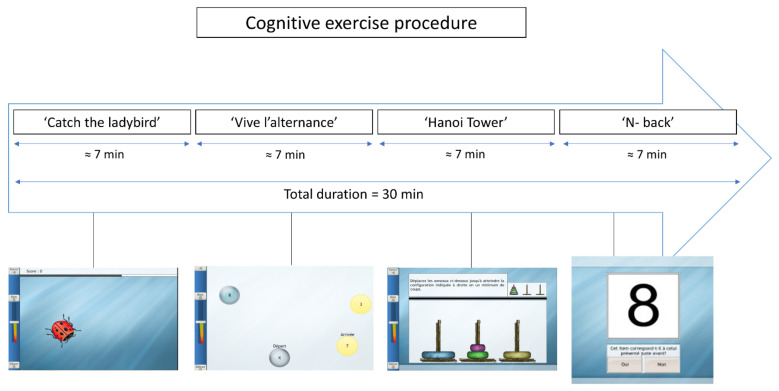
Illustration of cognitive exercise procedure.

**Figure 4 brainsci-12-00455-f004:**
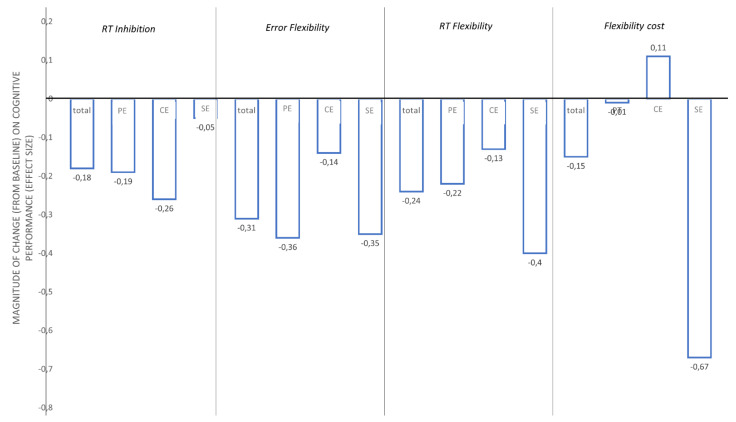
Magnitude of change (from baseline) on cognitive performance (Effect size). Negative effect size indicates a reduced reaction time and a lesser error produced after each intervention. PE: Physical Exercise; CE: Cognitive exercise; SE: simultaneous exercise.

**Figure 5 brainsci-12-00455-f005:**
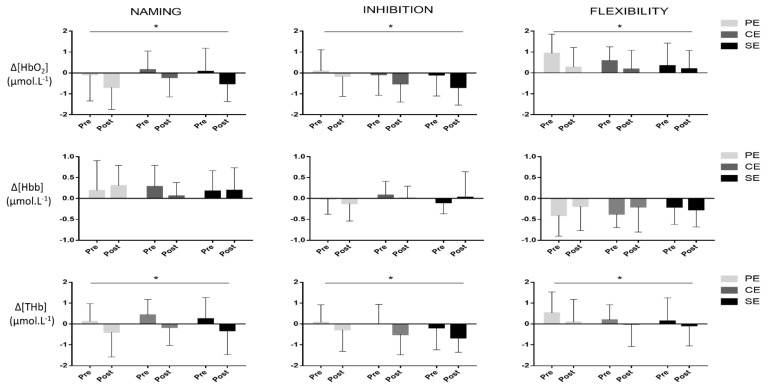
Cerebral oxygenation during Stroop test before and after intervention for the whole PFC (Prefrontal Cortex); * significant (*p* < 0.05). PE: Physical Exercise; CE: Cognitive Exercise; SE: simultaneous Exercise.

**Table 1 brainsci-12-00455-t001:** Anthropometric, blood pressure, and cognitive characteristics of participants.

Sex	Women	12
	Men	5
Age		62.4 ± 3.9
Height (cm)		166 ± 8.1
Weight (Kg)		69.0 ± 12.9
Arterial pressure (Mean of 3 measures)	Systolic	124.0 ± 13.8
	Diastolic	77.3 ± 11.4
MOCA		27.9 ± 1.7
BDI		3.0 ± 3.55

Results are presented: mean ± SD. BDI: Beck’s Depression Inventory. MoCA: Montreal Cognitive Assessment.

**Table 2 brainsci-12-00455-t002:** Cognitive exercise (game) used to perform cognitive training.

Name of The Cognitive Game	Testing Order	Cognitive Function	Aim of the Cognitive Exercise
Catch the ladybird	1	Processing speed	Press a ladybird as quickly as possible
‘Vive l’alternance’	2	Flexibility-Working memory	Based on the principle of the Trail Making Test Sort a list of words in alphabetical order and a list of numbers in ascending order, alternating between the two lists.
‘Hanoi Tower’	3	Planification	Principle of the Hanoi Tower test with different degrees of difficulty.
N-back	4	Working memory	Principle of the n-back test with different elements (numbers, shapes, colours) and different degrees of difficulty.

**Table 3 brainsci-12-00455-t003:** Cognitive performance before and after intervention.

		Overall		PE		CE		SE		ANOVA Analysis	
		Pre	Post	Pre	Post	Pre	Post	Pre	Post	Main Effect (*p* Value)	Interaction (*p* Value)
Naming	Errors (nb)	0.35 ± 0.62	0.50 ± 0.81	0.47 ± 0.71	0.53± 0.87	0.35 ± 0.61	0.69 ± 0.94	0.24 ± 0.56	0.30 ± 0.60	0.14	0.45
	RT (ms)	809.8 ± 121.3	796.8 ± 149.2	805.1 ± 133.7	798.1 ± 165.7	819.2 ± 119.8	811.6 ± 151.3	805.2 ± 116.7	781.5 ± 137.0	0.08	0.83
Inhibition	Errors (nb)	0.73 ± 1.8	0.36 ± 0.53	1.06 ± 2.9	0.24 ± 0.4	0.65 ± 1.1	0.44 ± 0.6	0.47 ± 0.9	0.41 ± 0.5	0.59	0.69
	RT (ms)	962.6 ± 197.5	926.8 ± 185.2	975.8 ± 231.7	928.6 ± 212.1	976.4 ± 185.6	926.6 ± 171.3	935.6 ± 180.3	925.3 ± 181.2	<0.01	0.23
Inhibition Cost	Errors (nb)	0.36 ± 1.7	−0.14 ± 0.9	0.58 ± 2.6	−0.29 ± 0.9	0.25 ± 1.1	−0.25 ± 1.2	0.23 ± 1.0	0.11 ± 0.8	0.24	0.61
	RT (ms)	156.4 ± 139.6	135.2 ± 107.5	174.5 ± 175.1	133.9 ± 112.3	160.8 ± 126.5	130.4 ± 100.4	134.0 ± 115.3	141.1 ± 115.1	0.70	0.59
Flexibility	Errors (nb)	4.0 ± 4.8	2.7 ± 3.2	4.1 ± 5.3	2.3 ± 2.8	4.1 ± 5.5	3.4 ± 4.0	3.7 ± 3.6	2.5 ± 2.6	0.01	0.83
	RT (ms)	1220.3 ± 210.5	1161.1± 222.5	1232.1 ± 200.7	1183.6 ± 211.4	1216.8 ± 199.3	1186.0 ± 241.8	1212.0 ± 241.2	1114.0 ± 218.9	<0.01	0.40
Flexibility Cost	Errors (nb)	3.3 ± 5.1	2.3 ± 3.1	3.1 ± 6.5	2.1 ± 2.6	3.5 ± 5.3	2.8 ± 4.0	3.2 ± 3.2	2.1 ± 2.6	0.06	0.98
	RT (ms)	285.7 ± 140.5	250.6 ± 154.8	281.5 ± 101.5	267.3 ± 104.8	276.7 ± 170.2	281.7 ± 205.8	298.9 ± 148.7	202.7 ± 133.3	0.01	0.17

Results are presented: mean ± standard deviation; RT: reaction time; PE: Physical Exercise; CE: Cognitive exercise; SE: simultaneous exercise; nb: number.

**Table 4 brainsci-12-00455-t004:** Cerebral oxygenation during Stroop test before and after intervention for the right and left hemisphere.

			Overall		PE		CE		SE		ANOVA Analysis	
			Pre	Post	Pre	Post	Pre	Post	Pre	Post	Main Effect (*p* Value)	Interaction (*p* Value)
Naming	Left	Δ[HbO_2_] (µmol·L^−1^)	0.04 ± 1.2	−0.53 ± 1.1	−0.04 ± 1.5	−0.80 ± 1.2	0.19 ± 0.99	−0.27 ± 1.1	−0.002 ± 1.2	−0.54 ± 0.86	**<0.01**	0.63
		Δ[Hbb] (µmol·L^−1^)	0.20 ± 0.58	0.17 ± 0.47	0.16 ± 0.75	0.25 ± 0.50	0.29 ± 0.53	0.05 ± 0.29	0.15 ± 0.45	0.20 ± 0.58	0.92	0.58
		Δ[THb] (µmol·L^−1^)	0.25 ± 1.0	−0.36 ± 1.2	0.12 ± 0.92	−0.54 ± 1.36	0.48 ± 0.94	−0.22 ± 1.0	0.15 ± 1.1	−0.33 ± 1.1	**<0.01**	0.89
	Right	Δ[HbO_2_] (µmol·L^−1^)	0.05 ± 1.0	−0.40 ± 0.94	−0.11 ± 1.1	−0.58 ± 0.98	0.11 ± 0.94	−0.16 ± 0.84	0.14 ± 1.1	−0.47 ± 0.99	**<0.01**	0.87
		Δ[Hbb] (µmol·L^−1^)	0.23 ± 0.67	0.20± 0.49	0.21 ± 0.87	0.35 ± 0.52	0.27 ± 0.58	0.07 ± 0.40	0.20 ± 0.58	0.18 ± 0.53	0.74	0.46
		Δ[THb] (µmol·L^−1^)	0.28 ± 0.99	−0.20 ± 1.1	0.11 ± 1.2	−0.23 ± 1.2	0.38 ± 0.78	−0.1 ± 0.85	0.34 ± 1.0	−0.28 ± 1.4	**<0.01**	0.91
Inhibition	Left	Δ[HbO_2_] (µmol·L^−1^)	−0.08 ± 1.0	−0.57 ± 1.0	0.14 ± 1.2	−0.28 ± 1.0	−0.13 ± 0.99	−0.59 ± 1.1	−0.24 ± 0.96	−0.82 ± 0.93	**0.02**	0.90
		Δ[Hbb] (µmol·L^−1^)	−0.05 ± 0.36	−0.05 ± 0.49	−0.01 ± 0.37	−0.15 ± 0.42	0.05 ± 0.40	0.05 ± 0.53	−0.18 ± 0.27	−0.06 ± 0.53	0.86	0.53
		Δ[THb] (µmol·L^−1^)	−0.13 ± 0.99	−0.62 ± 1.1	0.13 ± 1.0	−0.43 ± 1.01	−0.08 ± 0.96	−0.54 ± 1.4	−0.42 ± 0.95	−0.88 ± 0.71	**0.03**	0.88
	Right	Δ[HbO_2_] (µmol·L^−1^)	0.02 ± 1.0	−0.35 ± 0.87	0.03 ± 0.96	−0.04 ± 0.99	−0.03 ± 1.0	−0.43 ± 0.77	0.07 ± 1.2	−0.57 ± 0.81	**0.02**	0.50
		Δ[Hbb] (µmol·L^−1^)	0.03 ± 0.36	−0.001 ± 0.50	−0.01 ± 0.40	−0.09 ± 0.43	0.11 ± 0.31	−0.04 ± 0.26	−0.01 ± 0.39	0.12 ± 0.72	0.55	0.64
		Δ[THb] (µmol·L^−1^)	0.06 ± 1.0	−0.35 ± 0.87	0.02 ± 0.76	−0.13 ± 1.1	0.08 ± 0.99	−0.47 ± 0.75	0.06 ± 1.4	−0.45 ± 0.80	**0.02**	0.82
Flexibility	Left	Δ[HbO_2_] (µmol·L^−1^)	0.55 ± 1.0	0.12 ± 0.94	0.98 ± 1.1	0.18 ± 0.96	0.50 ± 0.73	0.11 ± 1.0	0.21 ± 1.1	0.07 ± 0.93	**<0.01**	0.06
		Δ[Hbb] (µmol·L^−1^)	−0.34 ± 0.46	−0.22 ± 0.69	−0.43 ± 0.49	−0.18 ± 0.62	−0.38 ± 0.41	−0.15 ± 0.92	−0.22 ± 0.49	−0.35 ± 0.47	0.54	0.65
		Δ[THb] (µmol·L^−1^)	0.21 ± 1.0	−0.1 ± 1.3	0.55 ± 1.0	−0.003 ± 1.2	0.12 ± 0.79	−0.04 ± 1.6	−0.01 ± 1.2	−0.28 ± 0.98	0.09	0.36
	Right	Δ[HbO_2_] (µmol·L^−1^)	0.66 ± 0.91	0.31 ± 0.91	0.89 ± 0.84	0.37 ± 1.0	0.65 ± 0.75	0.25 ± 0.90	0.47 ± 1.1	0.31 ± 0.89	**<0.01**	0.21
		Δ[Hbb] (µmol·L^−1^)	−0.31 ± 0.44	−0.20 ± 0.49	−0.37 ± 0.55	−0.19 ± 0.58	−0.37 ± 0.35	−0.25 ± 0.46	−0.18 ± 0.40	−0.18 ± 0.47	0.70	0.96
		Δ[THb] (µmol·L^−1^)	0.35 ± 1.0	0.10 ± 0.98	0.52 ± 1.1	0.18 ± 1.1	0.28 ± 0.87	0.01 ± 0.80	0.28 ± 1.1	0.12 ± 1.1	**0.03**	0.34

PE: Physical Exercise; CE: Cognitive exercise; SE: simultaneous exercise. Bold indicates significant results.

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
