# Peer review of "Acute Effect of a Simultaneous Exercise and Cognitive Task on Executive Functions and Prefrontal Cortex Oxygenation in Healthy Older Adults"

_brainsci, 2022, doi:10.3390/brainsci12040455_

Round 1

Reviewer 1 Report

The aim of this study was to investigate the effect of simultaneous physical and cognitive exercises on executive function in healthy older adults and performance was compared in comparison with either exercise or cognitive training alone. This is not a completely novel idea however, replication studies are needed in this area and the use of fNIRS adds a new element.

The article is generally well written, although there are occasional problems with grammar.

My main concern is that the purport conclusion of the study stated in the abstract is unsupported by the data. “A simultaneous physical and cognitive exercise are [an] efficient method than either training alone to improve executive function.” The results show that each of the three training conditions had similar effects on cognition and on the haemodynamic response. The claim is based on a non-significant interaction between time and group (p=0.12).

The presentation of changes in effect sizes in Figure 2 is not an intuitive way to present the data and leaves this reviewer unconvinced. Notably, as flexibility cost is calculated from the difference in flexibility RT and inhibition RT the resultant effect size for flexibility cost appears to be exaggerated for the ST condition when compared to the effect sizes of flexibility RT and inhibition RT separately. Error terms (which are not presented on this figure) are likely to be large given the complexity of the equation used to calculate the values presented. Thus claims of a ‘moderate’ effect aren’t convincing without knowing the error. Why not just present the data values for flexibly cost in the table above or in a figure? If flexibility cost is the key outcome on which the authors are making their claims, the reader needs to see descriptive data.

The authors need to add a limitations sections to their discussion to address why their hypothesis may not have been supported. Could more participants have led to different findings? The subject number is relatively small.

What was the time period between each of the conditions? Were these performed on separate days to allow any effects of the exercise to wash out? Were each of the tasks performed at a similar time of day?

Was the order of the 3 conditions randomized? A total of 1.5 hours of exercise seems fairly arduous for elderly people. Was each condition conducted on the same day or separate days. How were fatigue effects minimized?

The fNIRS findings of reduced activity in the DLPFC after exercise is unexpected. How were the haemodynamic concentrations changes determined, e.g. was this a mean overall value relative to a baseline period? The peak value? Area under the curve? Was the change in concentration calculated trial by trial during the Stroop task?

It states in the methods that “Initially resting PFC oxygenation was acquired in a seated position for 2 min before Stroop task.” Was this before EACH stroop task?

Was the fNIRS cap removed during the exercise phase of the protocol? Could differences in cap placement between the two cognitive measurements impacted the recordings?

 There is some evidence that anxiety can increased HbO levels (e.g. from walking studies (Holtzer et al., GeroScience 2019) ), could this have impacted the fNIRS recording in the initial assessment? Conceivably people post-exercise with a familiarity of the task may have been in a more relaxed state of mind.

Minor points:

How the physical and cognitive exercises are simultaneously combined is poorly described “a specific device (Velo-Cognitif, REV’LIM, Limoges, France) and the same modalities than previously described.” Please expand on this description to enable replication.

‘ST’, ‘CT’ and ‘PT’ are not defined in Figure 2, are these supposed to be ‘SE’, ‘CE’ and ‘PE’?

Line 148, missing word “older PEOPLE”

Provide the units for Table 3.

Abstract: the first line of the abstract is long and convoluted - please make it succinct.

Also, the second line appears incomplete: "To manage this socio-economic challenge [of an aging society] , physical, cognitive, and combined stimulation are proposed [to increase healthy independent living for longer].“

There is a mixture of referencing styles through the article. Eg. Sometimes references are numbered and other times the first author of the article is in parentheses. Eg Prince et al. is in the text, but doesn’t appear in the reference list. Attention to detail is required.

Please define what “MSSE” stands for. Perhaps the authors mean MMSE? The mini mental state examination? However MMSE are missing in Table 1, where MOCA scores are presented instead. Please clarify.

Line 101: “performed”

Line 148: incomplete sentence.

Line 196: Change “Strop” to “Stroop”

I was confused in the results the factor of ‘group’ is mentioned, although there is only one group of participants. Please define what ‘group’ is referring to in the statistical analysis section, perhaps ‘Condition’ or ‘Training’ would be more intuitive?

Author Response

Reviewer 1.

The aim of this study was to investigate the effect of simultaneous physical and cognitive exercises on executive function in healthy older adults and performance was compared in comparison with either exercise or cognitive training alone.

This is not a completely novel idea however, replication studies are needed in this area and the use of fNIRS adds a new element.

Response: Thank you for your comment. The topic of simultaneous cognitive and physical stimulation training on the cognitive functions of elderly people is indeed not new. Nevertheless there are very few data about the acute responseThe study by Ji et al, tested the effect of simultaneous stimulation of cognitive and physical exercise on cognitive function on the Stroop cognitive performance in the elderly. The cognitive exercise in their study used a single stimulus such as verbal fluency, which may explain why this study did not report a superior effect of dual stimulation on cognitive function in the elderly. Our study is the only one to our knowledge that used multiple modalities of cognitive exercise and their effect on cognitive functions of the elderly when they were implemented during physical exercise. We have added this information in the introduction to explain the novelty of our study.

The article is generally well written, although there are occasional problems with grammar.

Response: We have proofread the article and corrected some grammatical errors.

My main concern is that the purport conclusion of the study stated in the abstract is unsupported by the data. “A simultaneous physical and cognitive exercise are [an] efficient method than either training alone to improve executive function.” The results show that each of the three training conditions had similar effects on cognition and on the haemodynamic response. The claim is based on a non-significant interaction between time and group (p=0.12).

Response: We agree with you, and we have corrected this sentence in the abstract with the following sentence: "Simultaneous physical and cognitive exercise is as effective as either physical or cognitive exercise alone for improving executive function."

The presentation of changes in effect sizes in Figure 2 is not an intuitive way to present the data and leaves this reviewer unconvinced. Notably, as flexibility cost is calculated from the difference in flexibility RT and inhibition RT the resultant effect size for flexibility cost appears to be exaggerated for the ST condition when compared to the effect sizes of flexibility RT and inhibition RT separately.

Response: Thank you for your comment. We would like to keep this figure but we will add in the legend some explanatory elements to improve its understanding. Concerning the effect size of the cost of flexibility, we cannot assess it by comparing it to the effect size of the flexibility and inhibition RTs. In the light of our data, we have checked and it is not exaggerated.

Error terms (which are not presented on this figure) are likely to be large given the complexity of the equation used to calculate the values presented. Thus claims of a ‘moderate’ effect aren’t convincing without knowing the error. Why not just present the data values for flexibly cost in the table above or in a figure? If flexibility cost is the key outcome on which the authors are making their claims, the reader needs to see descriptive data.

Response: We have added the errors in the corresponding table.

The authors need to add a limitations sections to their discussion to address why their hypothesis may not have been supported. Could more participants have led to different findings? The subject number is relatively small.

 Response: We have added a limitations section to our article.

What was the time period between each of the conditions? Were these performed on separate days to allow any effects of the exercise to wash out? Were each of the tasks performed at a similar time of day?

Response: The 3 experimental conditions SE, CE and PE were carried out one week apart. The subjects always performed their sessions at the same time of the day. We will add this information in the article.

Was the order of the 3 conditions randomized? A total of 1.5 hours of exercise seems fairly arduous for elderly people. Was each condition conducted on the same day or separate days. How were fatigue effects minimized?

Response: As we said in your previous comment, the subjects performed the three 30-minute experimental conditions with a week's difference. And indeed the three experimental conditions were randomized between the subjects. We will check in the article that this information is clearly stated.

The fNIRS findings of reduced activity in the DLPFC after exercise is unexpected. How were the haemodynamic concentrations changes determined, e.g. was this a mean overall value relative to a baseline period? The peak value? Area under the curve? Was the change in concentration calculated trial by trial during the Stroop task?

Response: The results for fNIRS are not that unusual. Indeed, the vast majority of studies that have shown increased brain oxygenation during a cognitive task after physical exercise often use a baseline at the beginning of the exercise. In this sense, indeed, fNIRS results often suggest greater oxygenation due to increased cerebral blood flow and brain activation. In our case, we wanted to assess brain activation only and used a baseline before each cognitive task and before each block. The calculation of HBO2 and HHB values were calculated in relation to the baseline just before the blocks, by averaging the values between the beginning and the end of the block of the cognitive task. We have added this information in the article.

It states in the methods that “Initially resting PFC oxygenation was acquired in a seated position for 2 min before Stroop task.” Was this before EACH stroop task?

Response: Yes, as we said earlier, 2 minutes of rest ere implemented before each stroop test, before and after the experimental conditions.

Was the fNIRS cap removed during the exercise phase of the protocol? Could differences in cap placement between the two cognitive measurements impacted the recordings?

Response: We have placed the fNIRS optodes on the fp1 and fp2 regions of the 10/20 system, as specified above. For ease of placement, a pencil dot was drawn on the forehead for placement to ensure that the placement is always the same. It is therefore unlikely that the slight difference in placement, if any, would alter the results.

 There is some evidence that anxiety can increased HbO levels (e.g. from walking studies (Holtzer et al., GeroScience 2019) ), could this have impacted the fNIRS recording in the initial assessment? Conceivably people post-exercise with a familiarity of the task may have been in a more relaxed state of mind.

Response:  Indeed, as Holtzer shows, anxiety can modify the results of the fNIRS. We did not assess the anxiety of our subjects but by conducting an explanatory visit, familiarization with the stroop task and practice blocks before each experimental trial, could reduce a potential state of anxiety related to the experiment.

Minor points:

How the physical and cognitive exercises are simultaneously combined is poorly described “a specific device (Velo-Cognitif, REV’LIM, Limoges, France) and the same modalities than previously described.” Please expand on this description to enable replication.”

Response: We have added photos of the device and tables and photos of the cognitive training to improve the details of our experiment, in order to facilitate replication.

‘ST’, ‘CT’ and ‘PT’ are not defined in Figure 2, are these supposed to be ‘SE’, ‘CE’ and ‘PE’?

Response: We have corrected this mistake.

Line 148, missing word “older PEOPLE”

Response: We have corrected this mistake.

Provide the units for Table 3.

Response: We have added the units.

Abstract: the first line of the abstract is long and convoluted - please make it succinct.

Response: We have shortened the sentence.

Also, the second line appears incomplete: "To manage this socio-economic challenge [of an aging society] , physical, cognitive, and combined stimulation are proposed [to increase healthy independent living for longer].“

Response: On reading, this sentence seems complete with regard to the preceding sentence..

There is a mixture of referencing styles through the article. Eg. Sometimes references are numbered and other times the first author of the article is in parentheses. Eg Prince et al. is in the text, but doesn’t appear in the reference list. Attention to detail is required.

Response: We will check all the references in the article. Thank you

Please define what “MSSE” stands for. Perhaps the authors mean MMSE? The mini mental state examination? However MMSE are missing in Table 1, where MOCA scores are presented instead. Please clarify.

Response: We will clarify this mistake. Thank you.

Line 101: “performed”

Line 148: incomplete sentence.

Line 196: Change “Strop” to “Stroop”

Response: we have corrected accordingly.

I was confused in the results the factor of ‘group’ is mentioned, although there is only one group of participants. Please define what ‘group’ is referring to in the statistical analysis section, perhaps ‘Condition’ or ‘Training’ would be more intuitive?

Response: We have corrected accordingly.

Reviewer 2 Report

This is potentially an interesting study, but more work is needed before it is ready for publication.

For the Materials and Methods section, the heart rate data for 60% tHRmax during exercise should be shown. In addition, I don't really understand what the cognitive task of cognitive training is? Is there a difference between cognitive training task and Stroop task? I think you should describe it in more details. Furthermore, the authors mentioned a 30-min session of cognitive games, but what does each session mean? (Line 129) As for SE, I'm curious about simultaneous exercise and the cognitive task of how the operation was carried out and this needs to be described in order for the experiment to be reproducible.

In addition, the title of the Table 1 needs to be checked. I don't think the title of Figure 1 is appropriate that describe the Stroop task and the layout of the fNIRS sources and detectors. As for the SE (Simultaneously combined physical and cognitive exercise), is the cognitive training done throughout the exercise, are there breaks and does it last 30 minutes too long and what is the intensity of the exercise

The Results section is not clearly described. The data should be described more clearly in the Results section, instead of listing all the behavioral and fNIRS results through a table, the results are not focused. Visualizing fNIRS results might be better. The fNIRS results do not seem to support the conclusion that a simultaneous physical and cognitive exercise promotes executive function.

The Conclusion section is too long and not clear. I would suggest to revise it.

I also found that the references were not formatted correctly. (Line 40 and 61)

Line 101, performance? Should this be 'performed'?

Author Response

Reviewer 2.

This is potentially an interesting study, but more work is needed before it is ready for publication.

For the Materials and Methods section, the heart rate data for 60% tHRmax during exercise should be shown. In addition, I don't really understand what the cognitive task of cognitive training is? Is there a difference between cognitive training task and Stroop task? I think you should describe it in more details. Furthermore, the authors mentioned a 30-min session of cognitive games, but what does each session mean? (Line 129) As for SE, I'm curious about simultaneous exercise and the cognitive task of how the operation was carried out and this needs to be described in order for the experiment to be reproducible.

Response: Thank you for your comments. Regarding the first point, we have used heart rate as a measure to assess the intensity of the exercise but it is not a dependent variable so we do not present the evolution of the heart rate during the exercise.

In terms of cognitive training methodology, we have added information about the 'velo cognitif', the tasks used to perform the cognitive exercise. And indeed the Stroop task is not used as a cognitive training task but as a variable allowing us to assess the effect of our 3 experimental conditions on cognitive functions.

In addition, the title of the Table 1 needs to be checked. I don't think the title of Figure 1 is appropriate that describe the Stroop task and the layout of the fNIRS sources and detectors. As for the SE (Simultaneously combined physical and cognitive exercise), is the cognitive training done throughout the exercise, are there breaks and does it last 30 minutes too long and what is the intensity of the exercise

Response: The title of Table 1 and the title of the Figure 1 have been corrected. Concerning the SE, as we have now specified in the article, the cognitive training consisted of 4 cognitive games (Table 2 and Figure 3) and were carried out in the order presented during the 30 minutes. Between the cognitive game, 30 seconds passed in order to start the next game. As described in the manuscript, the intensity of the exercise is 60% of theorical maximal heart rate. All these information have been added in the manuscript.

The Results section is not clearly described. The data should be described more clearly in the Results section, instead of listing all the behavioral and fNIRS results through a table, the results are not focused. Visualizing fNIRS results might be better. The fNIRS results do not seem to support the conclusion that a simultaneous physical and cognitive exercise promotes executive function.

Response: In the results section all the figures and tables have been listed. We present the results in a specific order, first the cognitive data and then the fNIRS data. The results of the ANOVA are systematically reported and we believe that this section is clear. We will check this section for clarity. On the second point, we agree with you, and the fNIRS data suggest to us that the SE condition does not promote executive functions but this is also the case for the other conditions. Only our cognitive clinical data validate the fact that our 3 experimental conditions improve executive functions only.

The Conclusion section is too long and not clear. I would suggest to revise it.

Response: Thank you we have shortened the conclusion and revisited it.

I also found that the references were not formatted correctly. (Line 40 and 61)

Line 101, performance? Should this be 'performed'?

Response: We have corrected accordingly.

Round 2

Reviewer 2 Report

I have no further comments.